# Psychometric Properties of the Online International Physical Activity Questionnaire in College Students

**DOI:** 10.3390/ijerph192215380

**Published:** 2022-11-21

**Authors:** Marcus Vinicius Nascimento-Ferreira, Ana Clara Arrais Rosa, Jacyara Cristina Azevedo, Armando Rodrigues de Alencar Santos, Keisyanne De Araujo-Moura, Kelber Abrão Ferreira

**Affiliations:** 1Health, Physical Activity and Behavior Research (HEALTHY-BRA) Group, Universidade Federal do Tocantins, Miracema do Tocantins 77650-000, Brazil; 2Youth/Child Cardiovascular Risk and Environmental (YCARE) Research Group, Faculdade de Medicina, Universidade de Sao Paulo, São Paulo 01246-903, Brazil; 3Instituto de Ensino Superior do Sul do Maranhão (IESMA/UNISULMA), Imperatriz 65907-070, Brazil

**Keywords:** physical activity, psychometric properties, adults

## Abstract

Introduction: Due to the restrictions imposed to control the COVID-19 pandemic, there has been an increase in studies based on online surveys. However, there are important concerns about the validity and generalizability of results from online surveys. Thus, we aimed to test the reliability and validity of the online version of the International Physical Activity Questionnaire short form (IPAQ-SF) among college students from low-income regions. Methods: This was a methodological feasibility study with a random stratified sample from a college located in the state of Maranhão in the city of Imperatriz (Brazil). The sample consisted of 195 college students (at least 17 years of age) to evaluate the validity and 117 students to evaluate the reliability. All data were collected in a self-reported online format (via Google Forms) twice, with an interval of 2 weeks. We used Spearman’s correlation analysis for the reliability study. Additionally, we applied exploratory and confirmatory factor analysis to evaluate the structural validity. Results: The questionnaire showed acceptable (rho > 0.30) and significant (*p* < 0.05) reliability, except for the question about the duration of sitting time on a weekend day. When assessing the construct validity (exploratory analysis), we identified a single factor that explained 88.8% of the variance. The 1-factor model showed acceptable model fit (SRMR = 0.039; CFI = 0.96; TLI = 0.90) in confirmatory analysis. Conclusions: The online version of the IPAQ-SF has acceptable reliability among college students from low-income regions and maintains the structure of the construct regarding to physical activity.

## 1. Introduction

The COVID-19 pandemic led governments and health authorities to adopt strategies such as social distancing, restricting the movement of people and lockdowns [1]. These measures aggravated the global problem of physical inactivity [2] and reinforced the need for surveillance of physical activity in the population [3]. Previous studies have suggested that the level of physical inactivity exceeded 40% in several countries during the COVID-19 pandemic [3].

Subjective measurements of physical activity, such as questionnaires, recall, and diaries, are often economically and logistically viable alternatives in epidemiological studies [3,4]. The COVID-19 pandemic causes in-person research [5] to migrate to the online environment. The International Physical Activity Questionnaire short form (IPAQ-SF) is a popular tool that was developed and standardized globally for physical activity surveillance [6]. It has been used to monitor physical activity among adults in at least 12 countries [7].

Although the application of online tools may have a lower logistical cost and provide greater engagement than face-to-face data collection [1], some methodological challenges exist [1,5]. Differences in literacy and internet access can strongly distort participation and responsiveness to online subjective tools, especially in low- and middle-income countries or in societies with large differences in educational and socioeconomic levels [5]. In this sense, although the IPAQ-SF is recommended as a tool for monitoring physical activity [1], it has not yet been validated in the online format in low-income regions. Thus, our objective is to test the psychometric properties (reliability and validity) of the online version of the International Physical Activity Questionnaire short form (IPAQ-SF) among college students from low-income regions.

## 2. Materials and Methods

### 2.1. Study Design

This was a methodological study of reliability (temporal stability of responses) and construct validity (structural validity) [8] belonging to the first stage of a longitudinal observational multicenter project (cohort follow-up), entitled the 24 h movement behavior and metabolic syndrome (24 h-MESYN) study [9]. Data collection occurred during the first academic semester of 2021. A detailed information about 24 h-MESYN study can be found elsewhere [9].

### 2.2. Ethical Aspects

The project was approved by the Research Ethics Committee (CEP), opinion number: 4,055,604. This study followed the ethical principles for research with human beings: (i) Declaration of Helsinki, revised in 2008, Seoul, Korea; (ii) resolution of CNS 466/12; (iii) guidelines for the conduct of research activity during the COVID-19 pandemic (available at: http://www.fo.usp.br/wp-content/uploads/2020/07/Orientações-condução-de-pesquisas-e-atividades-CEP.pdf; accessed on 10 June 2020)); and (iv) guidelines for research in a virtual environment (OFÍCIO CIRCULAR N° 2/2021/CONEP/SECNS/MS). The students who agreed to participate in the study were provided information about the methodology of data collection and were given the chance to ask questions before voluntarily signing the informed consent form (online).

### 2.3. Population, Sample and Sampling

The population of this project was composed of students enrolled in a higher education institution in the city of Imperatriz (Maranhão, Brazil), which has a Gini index of 0.56 [10]. In 2020, the institution selected by convenience had 2225 students enrolled in 9 undergraduate programs (Administration, Law, Physical Education, Nursing, Aesthetics and Cosmetics, Physiotherapy, Nutrition, Psychology and Social Work).

The sample size was calculated according to the assumptions of Nascimento-Ferreira [11]. The parameters used to calculate the sample size were an α of 0.05, β of 0.10 (or power of 90%) and a Pearson correlation coefficient of 0.28 (for test–retest reliability). [4,11]. Based on these parameters, we estimated a required sample size of 98 students. Predicting losses of 50.0%, rejections of 50.0% and an incomplete data rate of 25.0% [12], we recruited 342 students to complete the survey at both timepoints (test–retest). The diversity of the sample was ensured via stratified random distribution [13] with respect to biological sex (at least 60.0% female), age (at least 25.0% up to 20 years) and study program (at least 60.0% in the health area) based on previous cohorts [14,15].

### 2.4. Inclusion and Exclusion Criteria

We included all regularly enrolled students, at least 17 years old, who were selected for the study and signed an informed consent form. We excluded students who did not complete or incorrectly completed the questionnaires. Students who exhibited physical disability or pregnancy were evaluated but were excluded from the analyses.

### 2.5. Procedures

The multidisciplinary fieldwork team consisted of undergraduate and graduate researchers in health programs. Mandatorily, the first contact (in person) with the participants occurred in the institution’s facilities and in the presence of graduate supervisors. In this first step, we explained the project and delivered the link via an instant messaging application (WhatsApp) with the informed consent form [9]. In this phase, we carried out the study invitation following national and regional health protocols and recommendations related to COVID-19 (e.g., wearing face mask, avoiding close contact). In the second step, after electronically signing the term, the participants answered the questionnaire (Q1, questionnaire first application) [9]. Here, our work team sent up to three reminders in the subsequent initial invitation (in the case of electronic questionnaire no response). In the third step, two weeks after the second step, we resent the questionnaire link, and the participants answered the questionnaire again (Q2, questionnaire second application) [9]. The questionnaire was sent to only those who replied to it in Q1. We adopted the same strategy of previous step for questionnaire reminders. In the latter two steps, the study contacts were restricted to messaging via WhatsApp.

Previously, the researchers participated in a training program with 20 h of work to obtain the necessary qualifications to perform the data collection [9]. The training was offered and supervised in the institution itself by scientists experienced in this type of study [16]. During training, we also reviewed the online version of the consent form and the questionnaires.

### 2.6. Study Variables

We accessed all the information via subjective instruments. These instruments provided operational measures for the following theoretical variables: biological sex, age, course nature, academic period and physical activity.

### 2.7. Instruments

We evaluated physical activity using the IPAQ-SF, a questionnaire validated for Brazilian adults [7]. This instrument measures physical activity via six questions about frequency and duration (of light [walking], moderate and vigorous physical activity) [7]. The questionnaire also assessed sitting time on weekdays and weekend days [7]. In addition, we retrieved sociodemographic (biological sex and age) and academic (course and academic shift) data. All information was self-reported and retrieved through an online questionnaire (available at https://forms.gle/L92wXsVaxxfPNgpE8 (accessed on 10 June 2020)).

### 2.8. Statistical Analysis

All statistical analyses were performed using Stata software version 15.0 (Stata Corporation, College Station, TX, USA). For all hypothesis tests, we established a criterion of statistical significance of 95.0% (*p* ≤ 0.05). The normality of the variables was assessed using the Shapiro-Wilk test. For descriptive analysis, continuous variables were described as medians and interquartile ranges. The categorical variables were described with absolute and relative frequencies. In the sensitivity evaluation, we verified the sample distributions between Q1 and Q2 with the chi-square goodness of fit test. To test reliability, Spearman’s correlation analysis was used with a cutoff ≥ 0.30 (for acceptable reliability) [17].

For the structural validity (construct validity), we adopted an exploratory factor analysis with varimax rotation, considering a factor loading of greater than 0.3 to retain items [13]. We extracted the factors based on the Kaiser rule, with eigenvalues greater than 1 necessary for factor retention [13]. Previously, we performed a preliminary analysis to determine whether the data were feasible with the Kaiser-Meyer-Olkin test (KMO > 0.50) for sample adequacy and the Bartlett test (*p* < 0.05 as statistically significant) for sphericity of the data [13]. Next, we adopted confirmatory factor analysis where we verified structure solution identified in the exploratory analysis. The fit indices to evaluate the quality of the model were, standardized root mean square residual (SRMR < 0.08), root mean square error of approximation (RMSEA < 0.08), Comparative Fit Index (CFI ≥ 0.90), Tucker-Lewis Index (TLI ≥ 0.90) [18].

## 3. Results

The demographic and academic characteristics are shown in Table 1. Of the 342 invited students, 57.0% completed the IPAQ-SF at the Q1 and 34.2% completed it at the Q2. For the reliability study, we used data from 117 participants (who completed the questionnaire at both applications). For the validity study, we used data from 195 participants (who completed the questionnaire at the first timepoint). At both questionnaire applications, the sample mainly consisted of female students aged between 21 and 25 years from the physical education course. We did not observe differential bias (*p* > 0.05) for demographic and academic characteristics.

We observed Spearman correlation coefficients ranging from 0.28 (for sedentary length of time on a weekend day) to 0.60 (for frequency of light physical activity) in the test–retest IPAQ-SF (Table 2). In the structural validity analysis, our data indicated viability for factoring (KMO = 0.682; Bartlett’s test, *p* < 0.001). Based on the exploratory factor analysis, we identified a single factor [labeled “habit of physical activity during the pandemic”] that explained 88.8% of the variance. Based on factor loading, the items that remained relevant for the construct were frequency of light, moderate and vigorous physical activity and duration of vigorous physical activity (Table 3). Further, we performed a confirmatory analysis on the factor model we identified (with 4 items). The 1-factor model exhibited an acceptable model fit (SRMR = 0.039; CFI = 0.96; TLI = 0.90), except for RMSEA of 0.149 (Table 4).

## 4. Discussion

During the pandemic, public health services and scientists are choosing to conduct data collection research remotely [1]. However, several challenges for remotely measuring behaviors related to movement exist, including recruitment and data quality [19]. The aim of the study was to test the reliability and validity of the online version of a large-scale international physical activity monitoring tool among college students from low-income regions. Our findings showed that the online version of the IPAQ-SF maintained its psychometric properties. Thus, the questionnaire can be a viable tool for monitoring physical activity in conditions of restricted social contact, as experienced in a pandemic context.

Although we adopted a face-to-face recruitment process and subsequent contacts via social network messages, our study showed a proportion of complete responses of 43.0% in the first timepoint, which decreased to 34.2% at the second timepoint. A high prevalence of data loss is frequently reported in health research conducted in Latin America [20] and in a similar population in Europe [15] and in Brazil [14]. The decrease in responses between timepoints may be attributed to decreases in the motivation participants to complete a second questionnaire after a short interval (two weeks). Additionally, recent studies have indicated that the socioeconomic and educational status of technological tools may be an important limiting factor for adherence to online research in the pandemic context [1,5]. Thus, we identified that a combination of approaches can increase the ability to recruit participants and collect data for monitoring physical activity in college students, such as (i) combining with snowball sampling [1], (ii) scheduling data collection from participants of the same class simultaneously [21]; (iii) enabling the student to participate via telephone from a colleague or relative [1], (iv) providing an SMS reminder [1,21] and face-to-face reminders [21].

Our findings showed that the online version of the IPAQ-SF had acceptably reliability, except for the question about the duration of sitting time (weekend day). These findings are in line with the literature [4,6], which indicates the robust reliability of this questionnaire as the main factor of its popularity for monitoring changes or trends in physical activity over time [6]. A comprehensive systematic review identified a wide variety of questionnaires, designed for different target populations and assessing different constructs and dimensions of physical activity in Brazilian population [22]. In this review, the scientists showed that the IPAQ was one of the most frequently investigated questionnaires [22]. Overall, the most reliable questionnaire was the internet version of Questionnaire of a Typical Physical Activity and Food Intake to youth population, whereas, IPAQ-SF achieved one of the best reliability performances (intraclass correlation coefficient > 0.70) for healthy adults [22]. Thus, our findigs regarding reliability of physical activity can be partially explained by the characteristics inherent to IPAQ to detect frequency and duration of a intensity of physicial activity (e.g., small number of items, well recognized activities) [23] and the format of administration we adopted (online delivering) with better performance in adults than other populations [22]. On the other hand, we speculate that the low level of reliability regarding the duration of sitting time on a weekend day can be attributed to the instability of this behavior [23] in college students compared to their weekly routines.

Additionally, the IPAQ-SF in the online format showed structural validity in our study, proving to be a tool capable of recovering the habit of physical activity (frequency and duration) but not for sitting time in a pandemic. The study of the structural validity of this questionnaire is not frequent in the literature, and its comparison with objective methods has been extensively reported in the last decade [4,6]. However, a study of construct validity in adult women observed that the IPAQ-SF is adequate to identify the level of physical activity through machine learning techniques [24]. Said that, the choice of the questionnaire should involve the physical activity domain (or type) of interest [23], which does not necessarily characterize the individual’s total physical activity level [22]. In this line, the IPAQ-SF included the four domains (leisure time physical activity, occupational activity, active travel and domestic activity) of physical activity [22,25] and sedentary behaviors [23,25]. A possible explanation for the preserved IPAQ structure in a pandemic could be the measurement of physical activity domains by different intensity questions (based on frequency and duration) avoiding absence or unbalanced responses if questions were asked by domain and/or specific activity.

In general, the IPAQ-SF is a tool that overestimates the total time of physical activity [6]; however, when only a subjective tool is accessible due to time and resource limitations for monitoring physical activity in adults, this questionnaire is recommended [4]. The scenario of limited economic and logistical resources is frequent in studies of college students in a low-income region [14], especially in a pandemic [1].

Our study has some limitations. Although our sample was robust in size and diversity, with proportions relative to age and biological sex similar to studies with representative samples [14,15], the results of this study cannot be extrapolated beyond the psychometric findings. Additionally, we observed a high proportion of nonparticipation, mitigated by the prediction (up to 50.0%) of losses/rejections in the study design. In post hoc analysis, the power of the sample (lowest correlation observed = 0.28; *n* = 117) remained significant (β = 0.13; power = 0.87). Another important aspect is that the research site was selected by convenience, and the sample was randomly selected. These choices are based on the sociodemographic, economic and academic diversity of the institution, which can provide us with a good idea of the characteristics of students from low-income regions, since representative methodological studies are not feasible [12] and ethically [11]. Furthermore, our sample was composed mainly by students from health sciences courses (~66.0%) which could reduce information bias; however, our sample compostion was similar to previous cohort study [14] in college students from Maranhão state (Brazil). Another factor that can be added is the level of education of the participants, as participants with a higher level of education have a greater likelihood of understanding the questions and therefore can provide more accurate answers, in contrast to the non-college adult population in a low-income region. Finally, the IPAQ-SF is a questionnaire and is susceptible to measurement errors inherent to subjective tools, such as memory bias and social desire [8].

On the other hand, this is the first study addressing the structural validity of the IPAQ-SF in the context of the pandemic in a low-income region. The challenges and lessons learned by collecting these data online are essential for the future implementation of these methods, which will likely become fundamental for the continuity of public health research [1] and the monitoring of physical activity [2].

## 5. Conclusions

The online version of the International Physical Activity Questionnaire short form presents acceptable reliability to measure the frequency and duration of walking, moderate activity and vigorous activity in college students and maintains the structure of the construct. However, the psychometric properties to support the use of the IPAQ-SF as an indicator of sitting time is not consistent. In the online format, the questionnaire offers a simple and low-cost alternative for monitoring the frequency and duration of physical activity in low-income regions.

## Figures and Tables

**Table 1 ijerph-19-15380-t001:** Sensitivity analysis according to demographic and academic variables in both questionnaires, first and second timepoint.

Variables	Q1 (*n* = 195)	Q2 (*n* = 117)	*p*-value ^†^
	%	%	
**Biological sex**			
Male	31.3	27.4	0.36
Female	68.7	72.6	
**Age**			
Up to 20 years	23.6	26.7	0.63
21 to 25 years	44.6	45.7	
26 to 30 years	18.5	14.7	
31 to 35 years	7.2	5.2	
36 years or older	6.2	7.8	
**Course**			
Nutrition	8.8	6.0	0.17
Physical Education	22.3	24.8	
Nursing	11.2	12.0	
Aesthetics and Cosmetics	7.6	1.7	
Physiotherapy	16.1	18.8	
Law	14.1	11.1	
Psychology	15.0	21.4	
Social work	3.3	17.7	
Administration	1.8	0.9	
**Shift**			0.92
Morning	20.1	20.5	
Evening	0.5	0.9	
Nocturnal	61.3	62.4	
Integral	18.1	16.2	

Q1, Questionnaire first application; Q2, Questionnaire second application. ^†^ Chi-square goodness of fit test. Significant values are in **bold** (*p* < 0.05).

**Table 2 ijerph-19-15380-t002:** Reliability analysis of the International Physical Activity Questionnaire short form (IPAQ-SF).

Variables		Q1	Q2	Rho
Light activity (walking)	Frequency (times/week)	2.0 (0.0–4.0)	1.0 (0.0–3.0)	**0.60**
	Duration (minutes/day)	20.0 (0.0–60.0)	0.0 (1.0–30.0)	**0.42**
Moderate activity	Frequency (times/week)	2.0 (0.0–5.0)	2.0 0.0–3.00	**0.59**
	Duration (minutes/day)	30 (0.0–60.0)	10.0 (0.0–60.0)	**0.32**
Vigorous activity	Frequency (times/week)	1.0 (0.0–4.0)	0.0 (0.0–3.0)	**0.57**
	Duration (minutes/day)	20 (0.0–60.0)	0.0 (0.0–60.0)	**0.30**
Sitting time	Duration of the week (minutes/day)	240 (60.0–360.0)	180 (48.0–360.0)	**0.47**
	Duration of the weekend (minutes/day)	180 (6.0–360.0)	120 (15.0–300.0)	**0.28**

Values are median (25th–75th percentile). Q1, Questionnaire first application; Q2, Questionnaire second application; rho, Spearman correlation coefficient. Significant values are in **bold** (*p* < 0.05).

**Table 3 ijerph-19-15380-t003:** Exploratory factor analysis of the International Physical Activity Questionnaire short form (IPAQ-SF).

Variables		Factor 1	Uniqueness	Communality (1-Uniqueness) %
Light Activity (walking)	Frequency (times/week)	0.615	0.468	53.2%
	Duration (minutes/day)		0.860	14.0%
Moderate activity	Frequency (times/week)	0.644	0.429	57.1%
	Duration (minutes/day)		0.705	29.5%
Vigorous activity	Frequency (times/week)	0.812	0.309	69.1%
	Duration (minutes/day)	0.516	0.502	49.8%
Sitting time	Duration of the week (minutes/day)		0.775	22.5%
	Duration of the weekend (minutes/day)		0.799	20.1%
**Eigenvalue (Proportion of Explained Variance)**		2.22 (0.88)		
**Cumulative Explained Variance ***		88.8%		

Factor loading < 0.30 was not showed. ***** Based on the factor identified via Eigenvalue > 1 (Kaiser rule).

**Table 4 ijerph-19-15380-t004:** Confirmatory factor analysis of the International Physical Activity Questionnaire short form (IPAQ-SF).

Fit Indices	Exploratory Analysis Factor Solution
Chi-square (df)	270.30 (6)
AIC	4459.76
SRMR	0.039
RMSEA	0.149
CFI	0.96
TLI	0.90

AIC, Akaike information criterion; SRMR, standardized root mean square residual (acepatable fit: <0.08); RMSEA, root mean square error of approximation (acceptable fit: <0.08); CFI, comparative fit index (acceptable fit: ≥0.9); TLI, Tucker–Lewis index (acceptable fit: ≥0.9).

## Data Availability

Not applicable.

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
