# Peer review of "Psychometric Properties of the Online International Physical Activity Questionnaire in College Students"

_ijerph, 2022, doi:10.3390/ijerph192215380_

Round 1
Reviewer 1 Report
The authors (AA) aim to test the reliability and validity of the online version of the International Physical Activity Questionnaire short form (IPAQ-SF) among college students from low-income regions. This is an article useful to increase our knowledge of the issue. Addressing all the issues below reported could make this manuscript eligible for the publication.
The title seems too strong and conclusive thus it should be improved softening the message in light of the study limitations.
Methods:
Lines 87, 213: I suggest that AA change from gender to sex.
Line 92: AA could delete “… or the informed consent form.” among exclusion criteria, being present among inclusion criteria.
Results:
Lines 144-145: Could be bias the higher presence of student from the physical education course? How did the authors solve it? I suggest that AA include this issue among study limitations.
Conclusions:
AA should soften their conclusions in light of the study limitations.
Author Response
The authors (AA) aim to test the reliability and validity of the online version of the International Physical Activity Questionnaire short form (IPAQ-SF) among college students from low-income regions. This is an article useful to increase our knowledge of the issue. Addressing all the issues below reported could make this manuscript eligible for the publication.
Dear reviewer, we would like to thank by time devoted to our work. We have read carefully your comments and suggestions. These recommendations helped us to improve our work substantially.
Please, find below our responses point-by-point.
The title seems too strong and conclusive thus it should be improved softening the message in light of the study limitations.
Thank you for your comments. We agree with your suggestion. We have now softened the manuscript title. Please see, TITLE (page 1, line 1): “Psychometric properties of the online International Physical Activity Questionnaire in college students”
Methods:
Lines 87, 213: I suggest that AA change from gender to sex.
Thank you for your comments. We have now improved the term consistency and language over the manuscript. Please see, METHODS section (page 3, line 88): “The sample size... The diversity of the sample was ensured via stratified random distribution[13] with respect to biological sex (at least 60.0% female), age (at least 25.0% up to 20 years) and study program (at least 60.0% in the health area) based on previous cohorts[14, 15].”; and, DISCUSSION section (page 8, line 243): “Our study has some limitations. Although our sample was robust in size and diversity, with proportions relative to age and biological sexsimilar to studies with representative samples[14, 15], the results of this study cannot be extrapolated beyond the psychometric findings. … social desire[8].”
Line 92: AA could delete “… or the informed consent form.” among exclusion criteria, being present among inclusion criteria.
Thank you for your comments. We removed the sentence. Please see, METHODS section (page 3, line 91): “We included all regularly enrolled students, at least 17 years old, who were selected for the study and signed an informed consent form. We excluded students who did not complete or incorrectly completed the questionnaires. Students who exhibited physical disability or pregnancy were evaluated but were excluded from the analyses.”
Results:
Lines 144-145: Could be bias the higher presence of student from the physical education course? How did the authors solve it? I suggest that AA include this issue among study limitations.
Thank you for your comments. We agree with your concern. We have now provided this information in the manuscript. Please see, DISCUSSION section (page 8, line 252): “Our study has some limitations…. Furthermore, our sample was composed mainly by students from health sciences courses (~66.0%) which could reduce information bias; however, our sample compostion was similar to previous cohort study[14] in college students from Maranhão state (Brazil). … social desire[8].”
Conclusions:
AA should soften their conclusions in light of the study limitations.
Thank you for your comments. We agree with your suggestion. We also added a confirmatory factor analysis in our manuscript in order to make our inferences more robust than the previous ones. However, we have now softened our inference over the manuscript. Please see, TITLE (page 1, line 1): “Psychometric properties of the online International Physical Activity Questionnaire in college students”; and, ABSTRACT section (page 1, line 28): Conclusion: “The online version of the IPAQ-SF has acceptable reliability among college students from low-income regions and maintains the structure of the construct regarding to physical activity.”; and, CONCLUSION section (page 9, line 267): “The online version of the International Physical Activity Questionnaire short form presents acceptable reliability to measure the frequency and duration of walking, moderate activity and vigorous activity in college students and maintains the structure of the construct. However, the psychometric properties to support the use of the IPAQ-SF as an indicator of sitting time is not consistent. In the online format, the questionnaire offers a simple and low-cost alternative for monitoring the frequency and duration of physical activity in low-income regions.”

Reviewer 2 Report
The paper by Nascimento-Ferreira et al titled "Validity of the online International Physical Activity Questionnaire in college students" reported test results on the reliability and validity of the online version of the International Physical Activity Questionnaire short form (IPAQ-SF) among college students from low-income regions.
On the whole, I think the manuscript is well written and the main conclusions are believable, I have only two minor suggestions for the authors.
1) It is not entirely clear when the data collection occurred (authors only said the sample was selected in 2020). This makes it hard to comprehend the broader temporal context. For instance, before COVID-19 would be a very different scenario compared to post-2019. The authors should make explicit the research design details.
2) The existing discussion is fairly weak, in the sense that it does not readily engage the broader literature on online survey data collection nor studies pertaining to data collection during pandemics. The authors could elaborate on the broader theoretical and practical implications.
Author Response
The paper by Nascimento-Ferreira et al titled "Validity of the online International Physical Activity Questionnaire in college students" reported test results on the reliability and validity of the online version of the International Physical Activity Questionnaire short form (IPAQ-SF) among college students from low-income regions. On the whole, I think the manuscript is well written and the main conclusions are believable, I have only two minor suggestions for the authors.
Dear reviewer, we would like to thank by time devoted to our work. We have read carefully your comments and suggestions. These recommendations helped us to improve our work substantially.
Please, find below our responses point-by-point.
1) It is not entirely clear when the data collection occurred (authors only said the sample was selected in 2020). This makes it hard to comprehend the broader temporal context. For instance, before COVID-19 would be a very different scenario compared to post-2019. The authors should make explicit the research design details.
Thank you for your comments. We agree with your suggestion. We also added information about study design adapted to pandemic period. Please see, METHODS section (page 3, line 100): “The multidisciplinary fieldwork team consisted of undergraduate and graduate researchers in health programs. Mandatorily, the first contact (in person) with the participants occurred in the institution's facilities and in the presence of graduate supervisors. In this first step, we explained the project and delivered the link via an instant messaging application (WhatsApp) with the informed consent form[9]. In this phase, we carried out the study invitation following national and regional health protocols and recommendations related to COVID-19 (e.g., wearing face mask, avoiding close contact). In the second step, after electronically signing the term, the participants answered the questionnaire (Q1, questionnaire first application)[9]. Here, our work team sent up to three reminders in the subsequent initial invitation (in the case of electronic questionnaire no response). In the third step, two weeks after the second step, we resent the questionnaire link, and the participants answered the questionnaire again (Q2, questionnaire second application)[9]. The questionnaire was sent to only those who replied to it in Q1. We adopted the same strategy of previous step for questionnaire reminders. In the latter two steps, the study contacts were restricted to messaging via WhatsApp.”
Please, also see some examples of our field work. Our work team in the first step (invitation phase):
Our work team in the second/third step (questionnaire or reminder sending):
2) The existing discussion is fairly weak, in the sense that it does not readily engage the broader literature on online survey data collection nor studies pertaining to data collection during pandemics. The authors could elaborate on the broader theoretical and practical implications.
Thank you for your comments. We agree with your suggestion (in line with reviewer #2 and #3). We have now improved our manuscript discussion. Please see, DISCUSSION section (page 1, line 1): “Our findings showed that the online version of the IPAQ-SF had acceptably reliability, except for the question about the duration of sitting time (weekend day). These findings are in line with the literature[4, 6], which indicates the robust reliability of this questionnaire as the main factor of its popularity for monitoring changes or trends in physical activity over time.[6] A comprehensive systematic review identified a wide variety of questionnaires, designed for different target populations and assessing different constructs and dimensions of physical activity in Brazilian population[22]. In this review, the scientists showed that the IPAQ was one of the most frequently investigated questionnaires[22]. Overall, the most reliable questionnaire was the internet version of Questionnaire of a Typical Physical Activity and Food Intake to youth population, whereas, IPAQ-SF achieved one of the best reliability performances (intraclass correlation coefficient > 0.70) for healthy adults[22]. Thus, our findigs regarding reliability of physical activity can be partially explained by the characteristics inherent to IPAQ to detect frequency and duration of a intensity of physicial activity (e.g. small number of items, well recognized activities)[23] and the format of administration we adopted (online delivering) with better performance in adults than other populations[22]. On the other hand, we speculate that the low level of reliability regarding the duration of sitting time on a weekend day can be attributed to the instability of this behavior [23] in college students compared to their weekly routines.
Additionally, the IPAQ-SF in the online format showed structural validity in our study, proving to be a tool capable of recovering the habit of physical activity (frequency and duration) but not for sitting time in a pandemic. The study of the structural validity of this questionnaire is not frequent in the literature, and its comparison with objective methods has been extensively reported in the last decade.[4, 6]. However, a study of construct validity in adult women observed that the IPAQ-SF is adequate to identify the level of physical activity through machine learning techniques.[24]. Said that, the choice of the questionnaire should involve the physical activity domain (or type) of interest[23], which does not necessarily characterize the individual’s total physical activity level[22]. In this line, the IPAQ-SF included the four domains (leisure time physical activity, occupational activity, active travel and domestic activity) of physical activity[22, 25] and sedentary behaviors[23, 25]. A possible explanation for the preserved IPAQ structure in a pandemic could be the measurement of physical activity domains by different intensity questions (based on frequency and duration) avoiding absence or unbalanced responses if questions were asked by domain and/or specific activity.”

Reviewer 3 Report
The article describes the validation of the International Physical Activity Questionnaire (IPAQ-SF) among university students from low-income regions of the city of Imperatriz in the state of Maranhão (Brazil). Based on a stratified random sample (according to biological sex and age), the reliability and validity of the online version (using google forms) of the aforementioned questionnaire was determined. Reliability was determined by Spearman's correlation analysis with a total of 117 participants who completed the questionnaire twice, with an interval of 2 weeks. The structural validity of the construct was determined from an exploratory factor analysis with 195 participants who completed the questionnaire only on the first occasion. The data obtained show acceptable (rho > 0.30) and significant (p < 0.05) reliability, except for the question on the duration of sitting time on a weekend day. The exploratory factor analysis reveals the existence of a single factor explaining 88.8% of the variance, which they call "physical activity habits during a pandemic". Following these results, the authors conclude that the online version of the IPAQ-SF has acceptable reliability among university students in low-income regions and maintains the construct structure.
The IPAQ questionnaire is a widely used instrument for determining physical activity status and has been validated in numerous populations from different countries and socio-economic backgrounds. The particular interest of the presented study is the validation of the online version of the questionnaire through Google forms, a widely used tool, especially at the height of the covid-19 pandemic. This tool is easy to use and allows data to be collected quickly and easily, so it is widely used and it is of interest to analyse both the reliability and validity of the responses obtained. On the other hand, the need and interest in the study and monitoring of the degree of physical activity in any population is unquestionable due to its repercussions on health, which increases the interest in having a tool for its determination and monitoring.
The authors adequately justify the importance and interest of the study in the introduction. In addition, the authors already establish some drawbacks of online tools, such as digital literacy and Internet access, which may affect both participation and the veracity of the responses.
While the procedure is well described, the results section is somewhat insufficient. It is not clear whether the 117 participants who completed the questionnaire on both occasions were the same, and whether they were among the 195 who completed the questionnaire only once. I consider that this question should be clarified as it affects the sample participating in the study.
Regarding the statistical tests used, they are adequate but could have been reinforced, especially in the analysis of the structural validity of the construct, with a Confirmatory Factor Analysis to establish whether a single factor is indeed established.
The discussion is adequate on the basis of the results obtained in the research and given that these are limited because it is a psychometric study, the discussion is also somewhat scarce as shown by the limited number of bibliographical references (a total of 20, given that reference 10 and 21 correspond to the same bibliographical reference).
The conclusion of the study is that the online version of the short-form International Physical Activity Questionnaire has acceptable reliability for measuring the frequency and duration of moderate and vigorous activity in university students and maintains the construct structure. In the online format, the questionnaire offers a simple and low-cost alternative for monitoring frequency and duration of physical activity in low-income regions.
The authors are aware of the limitations of their study but also of the opportunities presented by online tools despite their limitations.
Finally, I consider that the bibliographical references section should include a larger number of articles that support the studies carried out on this questionnaire (or even other online versions) in different populations to a greater extent. On the other hand, it should be revised as there is a duplication of citations, number 10 and 21, the latter being incomplete. The rest of the citations meet the standards set out in the journal.
Based on the above, it is recommended: Accept after minor revision (corrections to minor methodological errors and text editing)
Author Response
The article describes the validation of the International Physical Activity Questionnaire (IPAQ-SF) among university students from low-income regions of the city of Imperatriz in the state of Maranhão (Brazil). Based on a stratified random sample (according to biological sex and age), the reliability and validity of the online version (using google forms) of the aforementioned questionnaire was determined. Reliability was determined by Spearman's correlation analysis with a total of 117 participants who completed the questionnaire twice, with an interval of 2 weeks. The structural validity of the construct was determined from an exploratory factor analysis with 195 participants who completed the questionnaire only on the first occasion. The data obtained show acceptable (rho > 0.30) and significant (p < 0.05) reliability, except for the question on the duration of sitting time on a weekend day. The exploratory factor analysis reveals the existence of a single factor explaining 88.8% of the variance, which they call "physical activity habits during a pandemic". Following these results, the authors conclude that the online version of the IPAQ-SF has acceptable reliability among university students in low-income regions and maintains the construct structure.
Dear reviewer, we would like to thank by time devoted to our work.
The IPAQ questionnaire is a widely used instrument for determining physical activity status and has been validated in numerous populations from different countries and socio-economic backgrounds. The particular interest of the presented study is the validation of the online version of the questionnaire through Google forms, a widely used tool, especially at the height of the covid-19 pandemic. This tool is easy to use and allows data to be collected quickly and easily, so it is widely used and it is of interest to analyse both the reliability and validity of the responses obtained. On the other hand, the need and interest in the study and monitoring of the degree of physical activity in any population is unquestionable due to its repercussions on health, which increases the interest in having a tool for its determination and monitoring.
Dear reviewer, we would like to thank by time devoted to our work.
The authors adequately justify the importance and interest of the study in the introduction. In addition, the authors already establish some drawbacks of online tools, such as digital literacy and Internet access, which may affect both participation and the veracity of the responses.
We have read carefully your comments and suggestions. These recommendations helped us to improve our work substantially. Please, find below our responses point-by-point.
While the procedure is well described, the results section is somewhat insufficient. It is not clear whether the 117 participants who completed the questionnaire on both occasions were the same, and whether they were among the 195 who completed the questionnaire only once. I consider that this question should be clarified as it affects the sample participating in the study.
Thank you for your comments. We agree with your suggestion. We also added information about study design adapted to pandemic period. Please see, METHODS section (page 3, line 108): “The multidisciplinary fieldwork team consisted of undergraduate and graduate researchers in health programs. Mandatorily, the first contact (in person) with the participants occurred in the institution's facilities and in the presence of graduate supervisors. In this first step, we explained the project and delivered the link via an instant messaging application (WhatsApp) with the informed consent form[9]. In this phase, we carried out the study invitation following national and regional health protocols and recommendations related to COVID-19 (e.g., wearing face mask, avoiding close contact). In the second step, after electronically signing the term, the participants answered the questionnaire (Q1, questionnaire first application)[9]. Here, our work team sent up to three reminders in the subsequent initial invitation (in the case of electronic questionnaire no response). In the third step, two weeks after the second step, we resent the questionnaire link, and the participants answered the questionnaire again (Q2, questionnaire second application)[9]. The questionnaire was sent to only those who replied to it in Q1. We adopted the same strategy of previous step for questionnaire reminders. In the latter two steps, the study contacts were restricted to messaging via WhatsApp.”; and, RESULTS section (page 4, line 155): “The demographic and academic characteristics are shown in Table 1. Of the 342 invited students, 57.0% completed the IPAQ-SF at the Q1 and 34.2% completed it at the Q2. For the reliability study, we used data from 117 participants (who completed the questionnaire at both applications). For the validity study, we used data from 195 participants (who completed the questionnaire at the first timepoint). At both questionnaire applications, the sample mainly consisted of female students aged between 21 and 25 years from the physical education course. We did not observe differential bias (p> 0.05) for demographic and academic characteristics.”
Regarding the statistical tests used, they are adequate but could have been reinforced, especially in the analysis of the structural validity of the construct, with a Confirmatory Factor Analysis to establish whether a single factor is indeed established.
Thank you for your comments. We have now added confirmatory factor analysis in the manuscript. Please, see METHODS section (page 4, line 146): “For the structural validity (construct validity), we adopted an exploratory factor analysis with varimax rotation, considering a factor loading of greater than 0.3 to retain items[13]. We extracted the factors based on the Kaiser rule, with eigenvalues greater than 1 necessary for factor retention[13]. Previously, we performed a preliminary analysis to determine whether the data were feasible with the Kaiser‒Meyer‒Olkin test (KMO> 0.50) for sample adequacy and the Bartlett test (p <0.05 as statistically significant) for sphericity of the data[13]. Next, we adopted confirmatory factor analysis where we verified structure solution identified in the exploratory analysis. The fit indices to evaluate the quality of the model were, standardized root mean square residual (SRMR <0.08), root mean square error of approximation (RMSEA <0.08), Comparative Fit Index (CFI ≥0.90), Tucker‒Lewis Index (TLI ≥0.90)[18].”; and, RESULTS section (page 5, line 168): “We observed Spearman correlation coefficients ranging from 0.28 (for sedentary length of time on a weekend day) to 0.60 (for frequency of light physical activity) in the test-retest IPAQ-SF (Table 2). In the structural validity analysis, our data indicated via-bility for factoring (KMO = 0.682; Bartlett’s test, p <0.001). Based on the exploratory factor analysis, we identified a single factor [labeled “habit of physical activity during the pandemic”] that explained 88.8% of the variance. Based on factor loading, the items that remained relevant for the construct were frequency of light, moderate and vigorous physical activity and duration of vigorous physical activity (Table 3). Further, we performed a confirmatory analysis on the factor model we identified (with 4 items). The 1-factor model exhibited an acceptable model fit (SRMR = 0.039; CFI = 0.96; TLI = 0.90), except for RMSEA of 0.149 (Table 4).”; and, TABLE 4 (line 199):
The discussion is adequate on the basis of the results obtained in the research and given that these are limited because it is a psychometric study, the discussion is also somewhat scarce as shown by the limited number of bibliographical references (a total of 20, given that reference 10 and 21 correspond to the same bibliographical reference).
Thank you for your comments. We agree with your suggestion (in line with reviewer #1 and #3). We have now improved our manuscript discussion. Please see, DISCUSSION section (page 1, line 1): “Our findings showed that the online version of the IPAQ-SF had acceptably reliability, except for the question about the duration of sitting time (weekend day). These findings are in line with the literature[4, 6], which indicates the robust reliability of this questionnaire as the main factor of its popularity for monitoring changes or trends in physical activity over time.[6] A comprehensive systematic review identified a wide variety of questionnaires, designed for different target populations and assessing different constructs and dimensions of physical activity in Brazilian population[22]. In this review, the scientists showed that the IPAQ was one of the most frequently investigated questionnaires[22]. Overall, the most reliable questionnaire was the internet version of Questionnaire of a Typical Physical Activity and Food Intake to youth population, whereas, IPAQ-SF achieved one of the best reliability performances (intraclass correlation coefficient > 0.70) for healthy adults[22]. Thus, our findigs regarding reliability of physical activity can be partially explained by the characteristics inherent to IPAQ to detect frequency and duration of a intensity of physicial activity (e.g. small number of items, well recognized activities)[23] and the format of administration we adopted (online delivering) with better performance in adults than other populations[22]. On the other hand, we speculate that the low level of reliability regarding the duration of sitting time on a weekend day can be attributed to the instability of this behavior [23] in college students compared to their weekly routines.
Additionally, the IPAQ-SF in the online format showed structural validity in our study, proving to be a tool capable of recovering the habit of physical activity (frequency and duration) but not for sitting time in a pandemic. The study of the structural validity of this questionnaire is not frequent in the literature, and its comparison with objective methods has been extensively reported in the last decade.[4, 6]. However, a study of construct validity in adult women observed that the IPAQ-SF is adequate to identify the level of physical activity through machine learning techniques.[24]. Said that, the choice of the questionnaire should involve the physical activity domain (or type) of interest[23], which does not necessarily characterize the individual’s total physical activity level[22]. In this line, the IPAQ-SF included the four domains (leisure time physical activity, occupational activity, active travel and domestic activity) of physical activity[22, 25] and sedentary behaviors[23, 25]. A possible explanation for the preserved IPAQ structure in a pandemic could be the measurement of physical activity domains by different intensity questions (based on frequency and duration) avoiding absence or unbalanced responses if questions were asked by domain and/or specific activity.”
The conclusion of the study is that the online version of the short-form International Physical Activity Questionnaire has acceptable reliability for measuring the frequency and duration of moderate and vigorous activity in university students and maintains the construct structure. In the online format, the questionnaire offers a simple and low-cost alternative for monitoring frequency and duration of physical activity in low-income regions. The authors are aware of the limitations of their study but also of the opportunities presented by online tools despite their limitations.
Thank you for your comments. We agree with your suggestion (in line with reviewer #1). We also added a confirmatory factor analysis in our manuscript in order to make our inferences more robust than the previous ones. However, we have now softened our inference over the manuscript. Please see, TITLE (page 1, line 1): “Psychometric properties of the online International Physical Activity Questionnaire in college students”; and, ABSTRACT section (page 1, line 28): Conclusion: “The online version of the IPAQ-SF has acceptable reliability among college students from low-income regions and maintains the structure of the construct regarding to physical activity.”; and, CONCLUSION section (page 9, line 267): “The online version of the International Physical Activity Questionnaire short form presents acceptable reliability to measure the frequency and duration of walking, moderate activity and vigorous activity in college students and maintains the structure of the construct. However, the psychometric properties to support the use of the IPAQ-SF as an indicator of sitting time is not consistent. In the online format, the questionnaire offers a simple and low-cost alternative for monitoring the frequency and duration of physical activity in low-income regions.”
Finally, I consider that the bibliographical references section should include a larger number of articles that support the studies carried out on this questionnaire (or even other online versions) in different populations to a greater extent. On the other hand, it should be revised as there is a duplication of citations, number 10 and 21, the latter being incomplete. The rest of the citations meet the standards set out in the journal.
Thank you for your comments. We agree with your suggestion (in line with reviewer #1 and #2). We have now improved our manuscript discussion. Please see, DISCUSSION section (page 1, line 1): “Our findings showed that the online version of the IPAQ-SF had acceptably reliability, except for the question about the duration of sitting time (weekend day). These findings are in line with the literature[4, 6], which indicates the robust reliability of this questionnaire as the main factor of its popularity for monitoring changes or trends in physical activity over time.[6] A comprehensive systematic review identified a wide variety of questionnaires, designed for different target populations and assessing different constructs and dimensions of physical activity in Brazilian population[22]. In this review, the scientists showed that the IPAQ was one of the most frequently investigated questionnaires[22]. Overall, the most reliable questionnaire was the internet version of Questionnaire of a Typical Physical Activity and Food Intake to youth population, whereas, IPAQ-SF achieved one of the best reliability performances (intraclass correlation coefficient > 0.70) for healthy adults[22]. Thus, our findigs regarding reliability of physical activity can be partially explained by the characteristics inherent to IPAQ to detect frequency and duration of a intensity of physicial activity (e.g. small number of items, well recognized activities)[23] and the format of administration we adopted (online delivering) with better performance in adults than other populations[22]. On the other hand, we speculate that the low level of reliability regarding the duration of sitting time on a weekend day can be attributed to the instability of this behavior [23] in college students compared to their weekly routines.
Additionally, the IPAQ-SF in the online format showed structural validity in our study, proving to be a tool capable of recovering the habit of physical activity (frequency and duration) but not for sitting time in a pandemic. The study of the structural validity of this questionnaire is not frequent in the literature, and its comparison with objective methods has been extensively reported in the last decade.[4, 6]. However, a study of construct validity in adult women observed that the IPAQ-SF is adequate to identify the level of physical activity through machine learning techniques.[24]. Said that, the choice of the questionnaire should involve the physical activity domain (or type) of interest[23], which does not necessarily characterize the individual’s total physical activity level[22]. In this line, the IPAQ-SF included the four domains (leisure time physical activity, occupational activity, active travel and domestic activity) of physical activity[22, 25] and sedentary behaviors[23, 25]. A possible explanation for the preserved IPAQ structure in a pandemic could be the measurement of physical activity domains by different intensity questions (based on frequency and duration) avoiding absence or unbalanced responses if questions were asked by domain and/or specific activity.”
Based on the above, it is recommended: Accept after minor revision (corrections to minor methodological errors and text editing)
Dear reviewer, one more time, we would like to thank by time devoted to our work.

Round 2
Reviewer 2 Report
The revision is fine.